# Human C15orf39 Inhibits Inflammatory Response via PRMT2 in Human Microglial HMC3 Cell Line

**DOI:** 10.3390/ijms25116025

**Published:** 2024-05-30

**Authors:** Min Zhang, Yaqi Xu, Gaizhi Zhu, Qi Zeng, Ran Gao, Jinming Qiu, Wenting Su, Renxi Wang

**Affiliations:** 1Beijing Institute of Brain Disorders, Laboratory of Brain Disorders, Ministry of Science and Technology, Collaborative Innovation Center for Brain Disorders, Capital Medical University, Beijing 100069, China; zhangmin@mail.ccmu.edu.cn (M.Z.); yaqi.xu@mail.ccmu.edu.cn (Y.X.); 112020010272@ccmu.edu.cn (G.Z.); zengqi2022@mail.ccmu.edu.cn (Q.Z.); gaoran@mail.ccmu.edu.cn (R.G.); xqxxps@mail.ccmu.edu.cn (J.Q.); 2Laboratory for Clinical Medicine, Capital Medical University, Beijing 100069, China

**Keywords:** C15orf39, microglia, inflammation, PRMT2, NF-κB

## Abstract

Microglia-mediated inflammatory response is one key cause of many central nervous system diseases, like Alzheimer’s disease. We hypothesized that a novel C15orf39 (MAPK1 substrate) plays a critical role in the microglial inflammatory response. To confirm this hypothesis, we used lipopolysaccharide (LPS)-and interferon-gamma (IFN-γ)-induced human microglia HMC3 cells as a representative indicator of the microglial in vitro inflammatory response. We found that C15orf39 was down-regulated when interleukin-6 (IL-6) and tumor necrosis factor-α (TNFα) expression increased in LPS/IFN-γ-stimulated HMC3 cells. Once C15orf39 was overexpressed, IL-6 and TNFα expression were reduced in LPS/IFN-γ-stimulated HMC3 cells. In contrast, C15orf39 knockdown promoted IL-6 and TNFα expression in LPS/IFN-γ-stimulated HMC3 cells. These results suggest that C15orf39 is a suppressive factor in the microglial inflammatory response. Mechanistically, C15orf39 interacts with the cytoplasmic protein arginine methyltransferase 2 (PRMT2). Thus, we termed C15orf39 a PRMT2 interaction protein (PRMT2 IP). Furthermore, the interaction of C15orf39 and PRMT2 suppressed the activation of NF-κB signaling via the PRMT2-IκBα signaling axis, which then led to a reduction in transcription of the inflammatory factors IL6 and TNF-α. Under inflammatory conditions, NF-κBp65 was found to be activated and to suppress C15orf39 promoter activation, after which it canceled the suppressive effect of the C15orf39-PRMT2-IκBα signaling axis on IL-6 and TNFα transcriptional expression. In conclusion, our findings demonstrate that in a steady condition, the interaction of C15orf39 and PRMT2 stabilizes IκBα to inhibit IL-6 and TNFα expression by suppressing NF-κB signaling, which reversely suppresses C15orf39 transcription to enhance IL-6 and TNFα expression in the microglial inflammatory condition. Our study provides a clue as to the role of C15orf39 in microglia-mediated inflammation, suggesting the potential therapeutic efficacy of C15orf39 in some central nervous system diseases.

## 1. Introduction

Microglia migrate into the central nervous system from the yolk sac during embryogenesis [1]. They are brain-resident mononuclear phagocytes and maintain brain homeostasis by surveilling the environment [2,3]. They also actively respond to dangerous signals such as pathogens by producing proinflammatory cytokines, including interleukin-6 (IL-6) and tumor necrosis factor-α (TNFα) [4,5,6]. When responding to adverse signals, microglia migrate toward the inflammatory site to clear toxic substances [7]. Once the inflammation response is excessive and un-controlled, irreversible damage may appear in the host tissue [5,7]. Microglia-mediated inflammation has been shown to have a significant role in many central nervous system diseases, such as Alzheimer’s disease (AD) [8,9,10]. Microglia are involved in the pathogenesis of AD by releasing inflammatory mediators such as IL-6 and TNFα, complement components, chemokines, and free radicals, all of which are known to contribute to beta-amyloid (Aβ) production and accumulation [11]. Importantly, a vicious cycle of inflammation can develop between Aβ accumulation, activated microglia, and microglial inflammatory mediators, which enhance Aβ deposition and neuroinflammation [11]. Nonsteroidal anti-inflammatory drugs may be associated with a lower risk of developing AD, suggesting that future therapy may focus on inflammation-specific targeting in the AD brain [12]. By elucidating the molecular pathways that control microglial inflammation, researchers can gain deeper insights into the pathological processes underlying various neurological disorders, such as AD, Parkinson’s disease, multiple sclerosis, and traumatic brain injury.

Human C15orf39 is encoded by open reading frame 39 in chromosome 15. Human C15orf39 is expressed in various tissues, including the brain, according to the National Center for Biotechnology Information (NCBI) database [13]. In addition, C15orf39 is mainly expressed in the cytoplasm, although it can also be expressed in the nucleus, according to the GeneCards database [14]. However, its biological function in the brain is still un-known and remains to be elucidated. To date, two articles [15,16] are available in which C15orf39 was mentioned when studying other topics. One of these studies showed a significant downregulation of C15orf39 in degenerative nucleus pulposus cells [15]. Another study demonstrated that C15orf39 may be a novel substrate of mitogen-activated protein kinase 1 (MAPK1) [16]. Based on MAPK signaling, which is involved in proinflammatory cytokines secreted by microglia [17], we suggest that this novel molecule, C15orf39 plays a critical role in the microglial inflammatory response. To confirm our hypothesis, we used lipopolysaccharide (LPS)- and interferon gamma (IFN-γ)-induced human microglia HMC3 cells as an indicator of in vitro inflammatory responses of microglia. We found that C15orf39 inhibited LPS/IFN-γ-induced inflammatory cytokines, including IL-6 and TNFα, via regulation of protein arginine methyltransferase 2 (PRMT2)-mediated nuclear factor kappa beta (NF-κB) signaling pathways in human microglia HMC3 cells. Our study suggests an important role for C15orf39 in microglia-mediated inflammation and suggests that C15orf39 may be a potential therapeutic target in many central nervous system diseases.

## 2. Results

### 2.1. C15orf39 Was Down-Regulated in LPS/IFN-γ-Stimulated Human Microglia HMC3 Cells

A previous study demonstrated that LPS/IFN-γ markedly induces microglial inflammatory responses, including IL-6 and TNFα production, in HMC3 cells [6]. To assess the role of human C15orf39 in microglia-mediated inflammatory responses, we first used LPS/IFN-γ to stimulate HMC3 cells for 24 h. qPCR assay demonstrated that LPS/IFN-γ significantly promoted IL-6 and TNFα expression (Appendix A). Both our data and the previous study [6] suggested that LPS/IFN-γ could efficiently induce microglial inflammatory responses, such as IL-6 and TNFα production.

We found that the C15orf39 protein expression level significantly declined in LPS/IFN-γ stimulated HMC3 cells (Figure 1A,B). Further analysis revealed that C15orf39 protein expression levels had decreased in both the cytoplasm and nucleus (Figure 1C–E). Furthermore, this significant downregulation resulted from a reduction in C15orf39 transcriptional levels (Figure 1F). These results indicated that C15orf39 might be involved in LPS/IFN-γ-stimulated microglial inflammatory response.

### 2.2. C15orf39 Overexpression Repressed IL-6 and TNFα Expression in LPS/IFN-γ-Stimulated HMC3 Cells

Results from our experiments suggest that C15orf39 may play an important role in microglia-mediated inflammatory response. Thus, we first overexpressed C15orf39 by transfecting C15orf39 gene with a green fluorescent protein (GFP) tag into HMC3 cells. We found that C15orf39 protein with a GFP tag was expressed in HMC3 cells (Figure 2A). In addition, C15orf39 mRNA expression was also up-regulated (Figure 2B), whereas IL6 and TNFα mRNA and protein levels had decreased (Figure 2C–F). These results suggest that C15orf39 inhibits the expression of microglial inflammatory cytokines, including IL-6 and TNFα.

### 2.3. C15orf39 Knockdown Promoted IL-6 and TNFα Expression in LPS/IFN-γ-Stimulated HMC3 Cells

To further explore the role of C15orf39 in the microglia-mediated inflammatory response, we knocked down C15orf39 expression in HMC3 cells using C15orf39-specific siRNA. We found that C15orf39 protein (Figure 3A) and mRNA (Figure 3B) expression decreased in C15orf39-specific siRNA-transfected HMC3 cells. In contrast to a decrease in expression of C15orf39, knockdown of C15orf39 expression led to a significant exacerbation of IL6 and TNFα mRNA and protein expression induced by LPS/IFN-γ (Figure 3C–F). Collectively, our data suggest that C15orf39 is a negative regulator of microglia-mediated inflammatory responses, including IL6 and TNFα production.

### 2.4. C15orf39 Inhibited NF-κB Activation by Interacting with PRMT2

To further investigate the underlying mechanisms by which C15orf39 suppresses microglial inflammatory responses, we sequenced mRNA profiles to determine gene expression changed by C15orf39 silence. As expected, C15orf39 knockdown promoted IL-6 and TNFα mRNA expression in HMC3 cells (Figure 4A). Totally, 4016 genes were up-regulated, and 4016 genes were down-regulated in C15orf39-knocked down HMC3 cells (Figure 4B). Kyoto Encyclopedia of Genes and Genomes (KEGG) enrichment analysis showed that these genes were enriched in the NF-κB signaling pathway (Figure 4C). C15orf39 is discovered to interact with PRMT2 by searching UniProt website. In addition, some literature studies reported that PRMT2 interacts with the nuclear factor of kappa light chain gene enhancer in B-cells inhibitor alpha (IκBα) to inhibit NF-κB signaling pathway activation [18,19]. Therefore, we hypothesized that C15orf39 inhibits the inflammatory response via PRMT2. As expected, we found that C15orf39 interacted with PRMT2 (Figure 4D). Critically, C15orf39 overexpression up-regulated PRMT2 and IκBα expression and accordingly led to a reduction in NF-κBp65 phosphorylation (Figure 4E), whereas C15orf39 knockdown produced a reduction in PRMT2 and IκBα expression and accordingly promoted NF-κBp65 phosphorylation (Figure 4F). Collectively, our data suggest that C15orf39 interacts with PRMT2 to inhibit NF-κB pathway activation.

### 2.5. C15orf39 Inhibited Microglial Inflammatory Response via PRMT2

To further verify the role of PRMT2 in C15orf39-mediated inhibition of the inflammatory response, C15orf39-overexpressed HMC3 cells were transfected with PRMT2-specific siRNA. A western blot assay demonstrated that PRMT2 was knocked down in C15orf39-overexpressed HMC3 cells (Figure 5A). Compared to the control, PRMT2 knockdown reversed the inhibitory effect of C15orf39 on NF-κBp65 phosphorylation (Figure 5A). Accordingly, IL6 and TNFα mRNA and protein levels were also reversed (Figure 5B–E). Collectively, these data indicate that PRMT2 participates in the inhibitory effect of C15orf39 on microglial IL-6 and TNFα production via regulation of NF-κBp65 phosphorylation.

### 2.6. C15orf39 Inhibited NF-κB Signaling through Regulating Cytoplasmic PRMT2

To identify the specific region in PRMT2 that interacted with C15orf39, full-length PRMT2 and two truncation mutants were constructed as described previously [18] (Figure 6A) and transfected into HMC3 cells. The results demonstrated that the 96–433 domain-deleted mutant failed to interact with C15orf39 (Figure 6B). This suggests that this domain may play a crucial role in the interaction of C15orf39, and PRMT2. According to the GeneCards database, position 124 (lysine residue) of PRMT2 has been shown to undergo ubiquitination. Therefore, we asked whether C15orf39 could influence the stability of PRMT2 due to an effect on PRMT2 ubiquitination. To address this question, C15orf39-knocked-down or overexpressed HMC3 cells were stimulated for 30 min by LPS/IFN-γ and subjected to nucleoplasmic separation. After C15orf39 knockdown, PRMT2 levels decreased, and accordingly, NF-κBp65 was activated, whereas after C15orf39 overexpression, PRMT2 levels increased, and accordingly, NF-κBp65 was inhibited in the cytoplasm (Figure 6C,D). These results suggest that C15orf39 may inhibit PRMT2 ubiquitination and further inhibit NF-κB signaling in the cytoplasm.

### 2.7. NF-κBp65 Inhibited C15orf39 Transcription

Results from these experiments showed the transcription of C15orf39 mRNA involved in the LPS/IFN-γ-stimulated microglial inflammatory response. Thus, the DNA sequence within −2000 bp of the human C15orf39 promoter region was analyzed. We found that there are many motifs that bind with NF-κBp65 (Figure 7A). Our experiments showed that when NF-κBp65 signaling pathway was activated, C15orf39 expression was reduced (Figure 7B). Therefore, we speculate that NF-κBp65 may function as a suppressive transcription factor that can cause a reduction in C15orf39 transcription. To address this speculation, MA (a DNA-binding activity inhibitor of NF-κBp65) was used to pre-treat HMC3 cells for 3 h and then stimulated for 24 h with LPS/IFN-γ. Interestingly, we found that C15orf39 protein and mRNA levels had significantly increased (Figure 7C,D). Critically, a luciferase reporter assay showed that NF-κBp65 significantly decreased the activity of C15orf39 promoter (Figure 7E,F). These data indicate that NF-κBp65 negatively regulates C15orf39 transcription.

Taken together, our findings suggest that C15orf39 interacts with PRMT2 to inhibit microglial inflammatory responses by suppressing NF-κB signaling to enhance C15orf39 transcription (Figure 8).

## 3. Discussion

Nowadays, most proteins have been well characterized in terms of their biological functions. However, the functions of a few proteins still remain un-known [20,21]. C15orf39 is an un-characterized protein. In the present study, we found that C15orf39 interacted with PRMT2 and suppressed NF-κB-mediated IL-6 and TNFα production in the steady microglia, but NF-κBp65, induced LPS/IFN-γ and inhibited the suppressive effect of C15orf39 on microglial IL-6 and TNFα production. For the first time, we have identified an important biological role of C15orf39 in maintaining the microglial steady state.

We found that C15orf39 produced biological effects, especially in terms of maintaining the microglial steady state via an interaction with PRMT2. PRMT2 is located on human chromosome 21 q22.3 [22,23]. According to the NCBI database, PRMT2 and C15orf39 are widely expressed in many organs, including the brain. Our data demonstrated that PRMT2 and C15orf39 are expressed mainly in the steady-state microglia but not in activated microglia. A previous study shed light on the significant associations between PRMT2 and LPS-induced inflammation in macrophages [19]. In the present study, we showed that LPS/IFN-γ induced activation and translocation of NF-κB to the nucleus, in which it binds the promoters of its target genes [18], such as C15orf39, and suppresses C15orf39 transcription. Once C15orf39 expression declined, it relaxed the suppressive effect of PRMT2 on NF-κB, which allowed translocation of NF-κB to the nucleus. Thus, C15orf39 appears to be an upstream regulator for PRMT2-mediated suppressive effect on NF-κB signaling pathway activation in microglia cells. However, future studies are required to address how C15orf39 precisely regulates PRMT2 protein (such as through PRMT2 ubiquitination).

Our data suggest that C15orf39 affects the microglial inflammatory response by regulating PRMT2-mediated NF-κB signaling pathway. RelA (p65), RelB, c-Rel, p50, and p52 are five subunits of the NF-κB family that are capable of forming hetero- or homo-dimeric complexes [24,25]. In all un-stimulated nucleated cells, IκB, the inhibitor of NF-κB, binds NF-κB complexes and retains them in the cytoplasm [26]. Our study showed that C15orf39 enhanced the binding of IκB and NF-κB complexes by interacting with PRMT2. This process resulted in a lack of inflammatory responses in the steady-state microglia. Thus, our study suggests that C15orf39 is an upstream suppressor of NF-κB signaling pathway activation in the steady-state microglia.

Inflammatory stimuli typically induce NF-κB phosphorylation and subsequently translocate to the nucleus, where these stimuli target the interesting gene promoter of interest [25,27,28]. We found that the C15orf39 promoter region has multiple NF-κBp65 binding sites. Our experiments also demonstrated that NF-κBp65 can bind C15orf39 promoter and suppress its subsequent activation. Inflammatory stimuli produced a reduction in C15orf39 expression to further enhance NF-κB signaling pathway activation. Thus, our study suggests that NF-κB is an upstream suppressor of C15orf39 transcription in inflammatory microglia.

NF-κB plays an important role in regulating immune inflammatory responses by controlling multiple downstream target genes’ expression [29,30]. Our data showed that NF-κB suppressed C15orf39 expression by binding to its promoter. More and more evidence has demonstrated that the NF-κB pathway plays a critical role in neurodegenerative diseases and brain injury [31,32]. In our study, we found that C15orf39 interacted with PRMT2 to suppress NF-κB-mediated IL-6 and TNFα production in the steady microglia, but LPS/IFN-γ induced NF-κB to reverse the suppressive effect of C15orf39 on microglial IL-6 and TNFα production. In the future, we should confirm in vivo that overexpressing C15orf39 can control the detrimental effects of NF-κB signaling in brain injury conditions.

Our study has many limitations. We only used one type of human cell line, HMC3. We need to examine more cell lines to prove our results. We need in vivo data to confirm our results. Considering that it is difficult to use human samples, we will explore the function of the C15orf39 homolog (mouse 1700017 B05 Rik) in mice. In addition, we only tested the pro-inflammatory cytokines IL-6 and TNFα. Thus, we also need to evaluate other pathways, such as the nucleotide-binding domain, leucine-rich-containing pyrin domain-containing protein 3 (NLRP3), or Toll-like receptor (TLR), and other interleukins, such as IL-1.

In summary, for the first time, we found that in steady-state microglia, C15orf39 interacts with PRMT2 to inhibit NF-κB signaling, which reverses suppression of C15orf39 transcription in inflammatory microglia. In the future, we should explore more the molecular biological function of C15orf39 and provide some valuable suggestions for treating central nervous system diseases, such as AD, via overexpression of C15orf39.

## 4. Materials and Methods

### 4.1. Reagents and Antibodies

LPS and IFN-γ were purchased from Sigma-Aldrich Corporation, Saint Louis, MO, USA, and MA was purchased from MedChemExpress Corporation, Monmouth Junction, NJ, USA. C15orf39-EGFP-expressing pLVX-AcGFP1-N1 plasmids and PRMT2-FLAG-expressing pcDNA3.1 (+) plasmids were generated by General Biological Technology Corporation, Chuzhou, China. The plasmid pGL4.10 with C15orf39 luciferase reporter was generated by Obio Technology Corporation, Shanghai, China. C15orf39 (cat #: SIGS0012436-1)- and PRMT2 (cat #: SIGS0006713-1)-specific siRNA were purchased from Ruibo Biotechnology Corporation, Guangzhou, China. Anti-C15orf39 (cat #: PA5-65318) and anti-GFP (cat #: G10362) antibodies were purchased from Invitrogen Corporation, Carlsbad, CA, USA. Anti-PRMT2 (cat #: 66885-1-Ig), anti-Flag (cat #: 20543-1-AP), and anti-GAPDH (cat #: 60004-1-Ig) antibodies were purchased from Proteintech (Wuhan, China). Anti-phospho-NF-κBp65 (cat #: 3033 T), anti-NF-κBp65 (cat #: 8242 T), and anti-IκBα (cat #: 4814 T) antibodies were purchased from Cell Signaling Technology (Beverly, MA, USA). The anti-LaminB1 antibody (cat #: sc-374015) was purchased from Santa Cruz Biotechnology (Santa Cruz, CA, USA).

### 4.2. Cell Culture and Transfection

Human microglial HMC3 cells [33] were gifted by Professor Tianci Yang at Xiamen Clinical Laboratory Quality Control Center. HMC3 cells were cultured in modified Eagle’s medium (MEM) supplemented with 10% fetal bovine serum (FBS), 100 U/mL penicillin, and 100 μg/mL streptomycin in a 5% CO_2_ incubator at 37 °C. Under some conditions, HMC3 cells were stimulated with LPS (1 μg/mL) + IFN-γ (500 ng/mL) (LPS/IFN-γ) for 24 h. C15orf39-specific siRNAs were transfected into the cells, and cells were harvested 48 h after transfection for further analysis.

### 4.3. Immunofluorescence (IF)

The IF assay was previously described [34,35]. Briefly, a cell-attached patch was prepared, after which HMC3 cells were stimulated with LPS + IFN-γ (LPS/IFN-γ) for 24 h. The cells were washed, formalin-fixed, and perforated. Subsequently, 5% bovine serum albumin (BSA) was added to the cells, and the cells were then incubated for 1 h at room temperature (RT), followed by incubation with anti-C15orf39 antibody at 4 °C overnight. The cells were then washed. The fluorescence-conjugated secondary antibody was added to the cells and incubated in the dark for 1 h at RT. Finally, the cells were rinsed, incubated with DAPI (Invitrogen), and visualized by IF microscopy.

### 4.4. Quantitative Real-Time PCR (qPCR)

The qPCR assay was previously described [36,37]. Briefly, according to the manufacturer’s instructions, we used Trizol reagent (Thermo scientific, Waltham, MA, USA) to extract the total RNA in HMC3 cells. Total RNA was then reverse transcribed into cDNA using Evo-MLV RT Mix Kit (Accurate Biology Corporation, Changsha, China). The cDNA was then amplified with specific primers (Appendix A) using a SYBR Green Premix Pro Taq HS qPCR Kit III (Low Rox Plus, Accurate Biology Corporation, Changsha, China). Finally, data were normalized to β-actin (ACTIN), and results from the target group were expressed as a fold-change compared to the control group (set to 1).

### 4.5. Immunoprecipitation (IP)

The IP assay was previously described [38,39]. Briefly, IP lysis buffer (NP40, 1 mM phenyl methyl sulfonyl fluoride [PMSF], and a protease inhibitor cocktail) was added to human HMC3 cells, after which cells were lysed for 30 min on ice. Cells were then centrifuged for 20 min at 12,000 rpm. An anti-GFP antibody was incubated with protein A-conjugated Dynabeads for 30 min at room temperature (RT). The mixture of anti-GFP antibody-protein A-conjugated Dynabeads was added to the HMC3 cell lysate supernatant and incubated for 1 h at RT. The IP wash buffer was used to wash the immunoprecipitates five times. Finally, a magnet was used to separate immunoprecipitates by removing the supernatant and resuspending them by gentle pipetting. Sodium dodecyl sulfate polyacrylamide gel electrophoresis (SDS-PAGE) loading buffer was used to elute the immunoprecipitated proteins to be used for western blotting.

### 4.6. Immunoblotting (IB)

The IB assay was previously described [40,41]. Briefly, western blots were used to determine the levels of the proteins of interest in HMC3 cells. Cells were first lysed with RIPA buffer (Solarbio Corporation, Beijing, China) supplemented with protease, 1 mM PMSF, and phosphatase inhibitor cocktails (Beyotime Corporation, Shanghai, China). The extracted proteins were separated by 10% SDS-PAGE and transferred to polyvinyl difluoride (PVDF) membranes (Millipore, Sigma Corporation, St. Louis, MO, USA). Tris-buffered saline with Tween (TBST) and 5% skim milk were used to block the PVDF membranes. PVDF membranes were washed three times and incubated with primary antibodies overnight at 4 °C. The primary antibodies include C15orf39 (Invitrogen), PRMT2 (Proteintech Group Inc., Rosemont, IL, USA), p-p65/p65 (Cell Signaling Technology, Inc., Beverly, MA, USA), IκBα (Cell Signaling Technology, Inc., Beverly, MA, USA), and glyceraldehyde 3-phosphate dehydrogenase (GAPDH) (Proteintech Group Inc., Rosemont, IL, USA). PVDF membranes were washed five times and incubated with a second antibody, such as horseradish peroxidase (HRP)-conjugated goat anti-mouse or goat anti-rabbit antibodies (Gene-Protein Link Biological Technology Corporation, Beijing, China) for 1 h. Finally, a substrate or excitation luminescence solution was added to PVDF membranes for color development to observe the presence or absence of the target band.

### 4.7. Enzyme-Linked Immunosorbent Assay (ELISA)

The ELISA assay was previously described [42,43]. Briefly, supernatants were collected from cultured HMC3 cells. According to the manufacturer’s instructions, the levels of TNF-α and IL-6 in supernatants were determined using commercially available ELISA kits (Invitrogen, CA, USA). Simply put, antibodies were added to a 96-well plate overnight at 4 °C. The plate was washed three times and dried by throwing. The supernatants were added to the antibody-coated plate and incubated for 30 min at 37 °C. The plate was washed three times and dried. Enzyme-linked antibody was added to the plate and incubated for 30 min at 37 °C. The plate was washed three times and dried. Substrate solution was added to the plate for 15 min at 37 °C, after which termination solution was added. An ELISA detector was used to measure the optical density (OD) value.

### 4.8. Extraction of Nuclear and Cytoplasmic Protein

A nuclear and cytoplasmic protein extraction assay was previously described [44]. Briefly, according to the manufacturer’s protocol, a mammalian nuclear and cytoplasmic protein extraction kit (TransGen, Beijing, China) was used to extract the nuclear and cytoplasmic proteins. Pre-treated HMC3 cells were washed twice with ice-cold phosphate-buffered saline (PBS). Ice-cold cell lysis buffer supplemented with cytoplasmic protein extraction buffer I (CPEBI)/II (CPEBII) was used to lyse cells. In one tube, cytoplasmic extracting solutions were used to gather cytoplasmic protein. In the other tube, nuclear protein extraction buffer (NPEB) was used to collect nuclear protein. Finally, nuclear [26] and cytoplasmic proteins were subjected to 10% SDS–PAGE and immunoblotting analysis.

### 4.9. Dual-Luciferase Reporter Assay System

The dual-luciferase reporter assay was previously described [45,46]. Briefly, the human C15orf39 promoter (−2000 bp to +228 bp) was generated by Obio Technology (Shanghai, China) and cloned into a pGL4.10 luciferase reporter vector. pGL4.10 with C15orf39 promoter, the control pRL-CMV-Renilla plasmid, and NF-κBp65-Flag-expressing or empty (vector) pcDNA3.1 plasmids were co-transfected into HMC3 cells. Seventy-two hours after transfection, the cells were lysed, and luciferase activity was measured.

### 4.10. Statistics

Statistical analyses were performed using ImageJ software (version: 1.8.0) and GraphPad Prism (version: 6, GraphPad Software Corporation, San Diego, CA, USA). The data are presented as the mean ± SD. Student’s *t*-test was used to compare two groups. *p* < 0.05 was considered statistically significant.

## Figures and Tables

**Figure 1 ijms-25-06025-f001:**
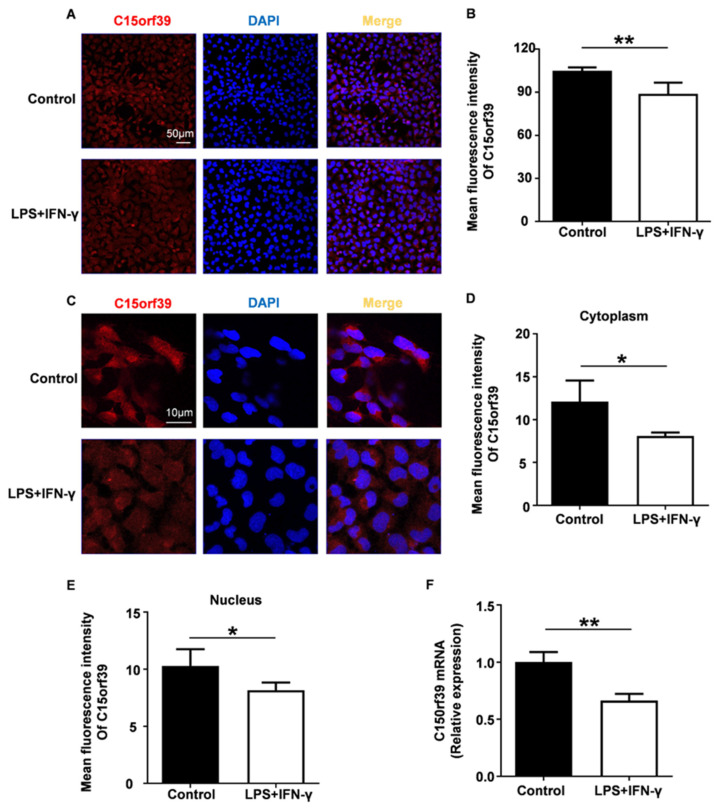
C15orf39 expression was down-regulated by lipopolysaccharide/interferon gamma (LPS/IFN-γ) stimulation in the human microglial clone 3 cell line (HMC3). (**A**,**B**) Immunofluorescent staining (**A**) and quantitative expression (**B**) of C15orf39 were analyzed in HMC3 cells treated with LPS/IFN-γ for 24 h. Red, anti-C15orf39 antibody staining; blue, 4′,6-diamidino-2-phenyliindole (DAPI) staining; scalebars, 50 μm. (**C**–**E**) Immunofluorescence staining (**C**) and quantitative expression (**D**,**E**) of C15orf39 in the cytoplasm (**D**) and nucleus (**E**) were analyzed in HMC3 cells treated with LPS/IFN-γ for 24 h. Red, anti-C15orf39 antibody staining; blue, DAPI staining; scalebars, 10 μm. (**F**) C15orf39 mRNA expression was analyzed using a real-time polymerase chain reaction (qPCR) assay in HMC3 cells treated with LPS/IFN-γ for 24 h. (**B**,**D**–**F**) Data indicate means ± standard deviation (SD), *n* = 3, two tailed Student’s *t*-test, * *p* < 0.05, ** *p* < 0.01.

**Figure 2 ijms-25-06025-f002:**
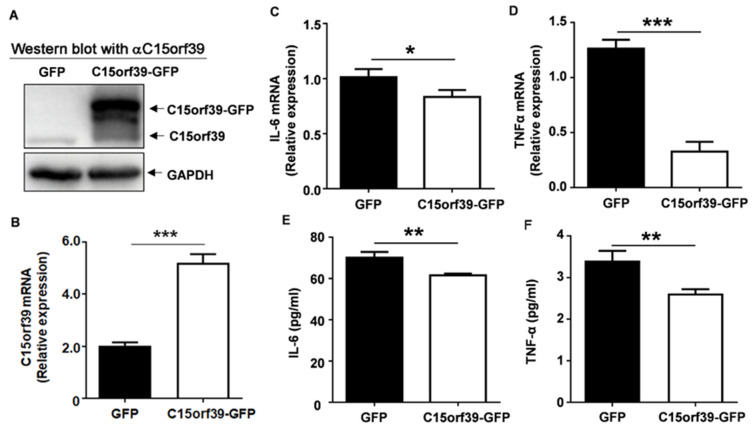
C15orf39 overexpression caused a reduction in interleukin 6 (IL-6) and tumor necrosis factor alpha (TNFα) expression induced by LPS/IFN-γ in HMC3 cells. HMC3 cells were transfected with C15orf39-GFP- or GFP-expressing plasmids for 72 h. Several experiments were performed. (**A**,**B**) C15orf39 protein (**A**) and mRNA (**B**) expression were determined by western blot and qPCR assay, respectively (*n* = 3). (**C**,**D**) IL-6 (**C**) and TNFα (**D**) mRNA expression were analyzed using qPCR. (**E**,**F**) IL-6 (**E**) and TNFα (**F**) protein expression was determined using an enzyme-linked immunosorbent assay (ELISA) in the cell supernatant. (**B**–**F**) Data indicate means ± SD, *n* = 3, two tailed Student’s *t*-test, * *p* < 0.05, ** *p* < 0.01, *** *p* < 0.0001.

**Figure 3 ijms-25-06025-f003:**
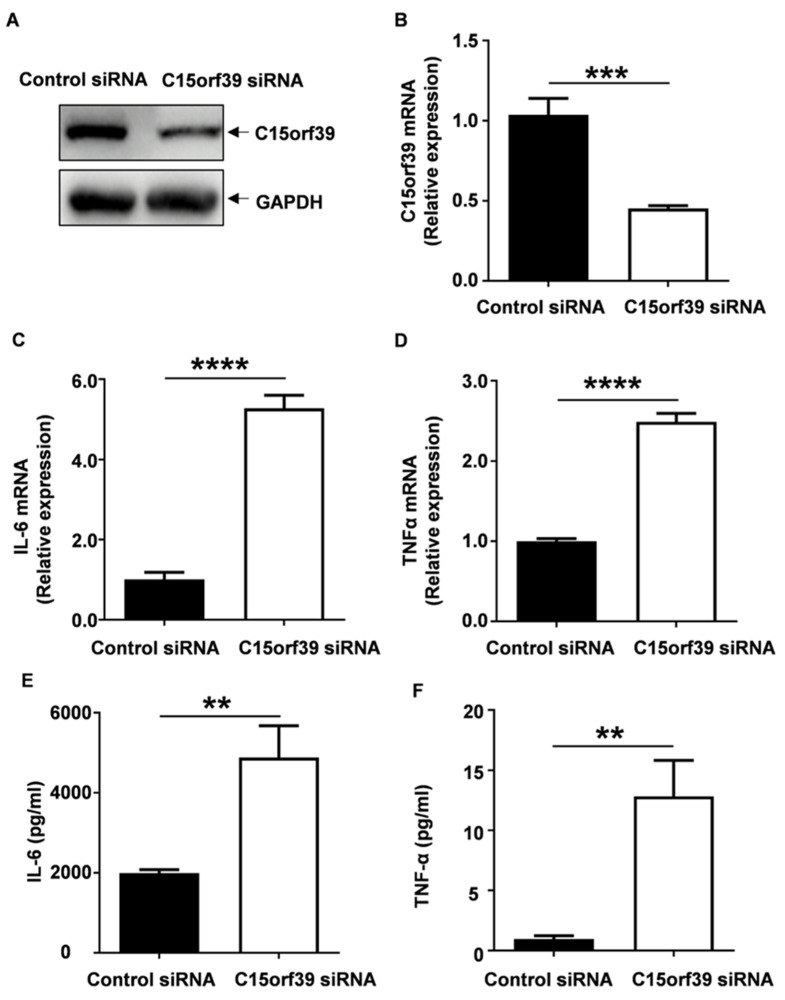
C15orf39 knockdown promoted IL-6 and TNFα expression induced by LPS/IFN-γ in HMC3. HMC3 cells were transfected with C15orf39 or control siRNA and then stimulated by LPS/IFN-γ for 24 h. Several experiments were performed. (**A**,**B**) C15orf39 protein (**A**) and mRNA (**B**) expression were determined by western blot and qPCR assay, respectively (*n* = 3). (**C**,**D**) IL-6 (**C**) and TNFα (**D**) mRNA expression were analyzed using qPCR. (**E**,**F**) IL-6 (**E**) and TNFα (**F**) protein expression was determined using ELISA assay in the cell supernatant. (**B**–**F**) Data indicate means ± SD, *n* = 3, two tailed Student’s *t*-test, ** *p* < 0.01, *** *p* < 0.001, **** *p* < 0.0001.

**Figure 4 ijms-25-06025-f004:**
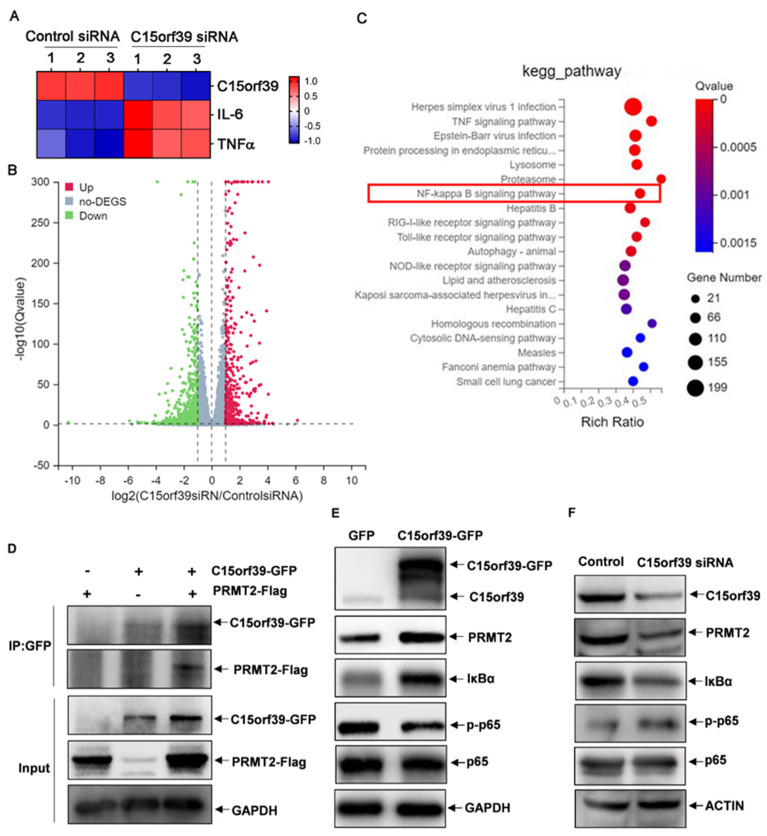
C15orf39 inhibited NF-κB signaling and promoted C15orf39-interacting PRMT2 expression. (**A**–**C**) HMC3 cells were transfected with C15orf39 or control siRNA and then stimulated by LPS/IFN-γ for 24 h. The cells were collected, and the transcripts in these cells were analyzed using RNA-sequencing (*n* = 3). Heatmap analysis of IL6 and TNF expression (**A**), volcano plot of the differential transcripts (upregulated genes number: 4016; down-regulated genes number: 4016) (**B**), and Kyoto Encyclopedia of Genes and Genomes (KEGG) enrichment analysis of the gene set (**C**) were analyzed and shown. (**D**) HMC3 cells were transfected with C15orf39-GFP and protein methyl arginine transferase 2 (PRMT2)-Flag. Cell lysates were immunoprecipitated with anti-GFP antibody, after which C15orf39 and PRMT2 expression were detected by western blot assays with anti-GFP anti-body and anti-Flag antibody (*n* = 3). (**E**) Western blot analysis of PRMT2, nuclear factor of kappa light chain gene enhancer in B-cells inhibitor alpha (IκBα) and the phosphorylated and total nuclear factor kappa beta (NF-κB) p65 levels in GFP- and C15orf39-GFP-overexpressed HMC3 cells (*n* = 3). (**F**) Western blot analysis of PRMT2, IκBα, and phosphorylated and total NF-κBp65 levels in control- and C15orf39-knocked down HMC3 cells that were stimulated with LPS/IFN-γ for 24 h (*n* = 3).

**Figure 5 ijms-25-06025-f005:**
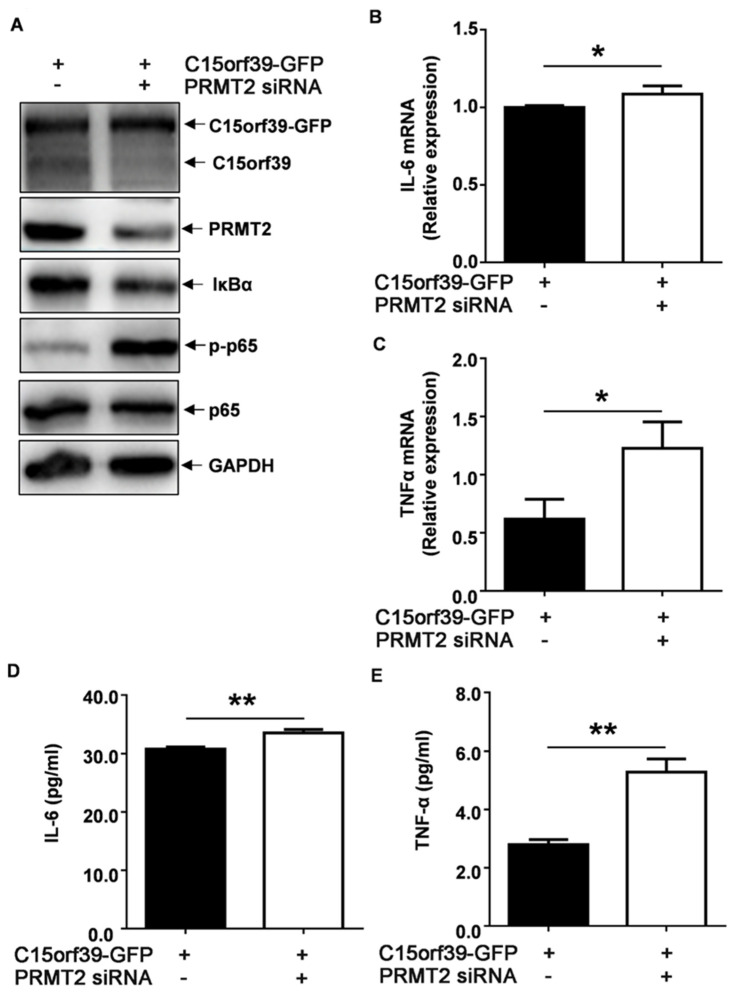
C15orf39 inhibited IL-6 and TNFα expression via PRMT2 signaling in LPS/IFN-γ-stimulated HMC3. HMC3 cells were transfected with C15orf39-GFP-expressing plasmids and control or PRMT2 siRNA for 72 h. Several experiments were performed. (**A**) C15orf39, PRMT2, IκBα, and phosphorylated and total NF-κBp65 protein levels were analyzed by western blot assay (*n* = 3). (**B**,**C**) IL-6 (**B**) and TNFα (**C**) mRNA expression were analyzed by qPCR assay. (**D**,**E**) IL-6 (**D**) and TNFα (**E**) protein levels in the cell supernatant were determined by ELISA. (**B**–**E**) Data indicate means ± SD, *n* = 3, two tailed Student’s *t*-test, * *p* < 0.05, ** *p* < 0.01.

**Figure 6 ijms-25-06025-f006:**
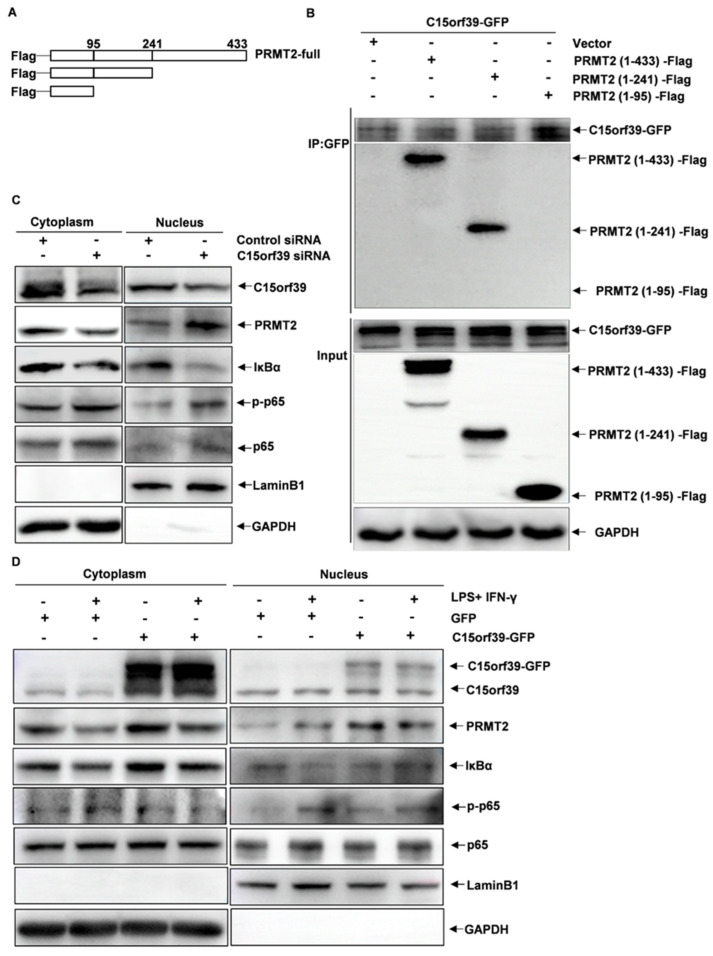
C15orf39 inhibited NF-κB signaling through regulation of cytoplasmic PRMT2. (**A**) Schematic diagram of the Flag-PRMT2 structural domain deletion construct. Upper panel: full-length PRMT2 (1–433); middle panel: PRMT2 (1–241) deleted C-terminal of PRMT2; lower panel: PRMT2 (1–95) deleted middle and C-terminal of PRMT2. (**B**) HMC3 cells were co-transfected with C15orf39-GFP-expressing plasmids and plasmids expressing PRMT2 structural domains [PRMT2 (1–433), PRMT2 (1–241), or PRMT2 (1–95)]. After performing co-immunoprecipitation (co-IP) with an anti-GFP antibody, structural domain-deleted PRMT2 was detected by western blotting using an anti-Flag antibody (*n* = 3). (**C**) HMC3 cells were transfected with control or C15orf39 siRNA for 72 h followed by nucleocytoplasmic separation. The protein levels of C15orf39, PRMT2, IκBα, and the phosphorylated and total NF-κBp65 were analyzed by western blot assay (*n* = 3). (**D**) HMC3 cells were transfected with GFP- or C15orf39-GFP-expressing pLVX-AcGFP1-N1 plasmids for 72 h and then stimulated with LPS/IFN-γ for 30 min, followed by nucleocytoplasmic separation. The protein levels of C15orf39, PRMT2, IκBα, and phosphorylated and total NF-κBp65 were analyzed by western blot assay (*n* = 3).

**Figure 7 ijms-25-06025-f007:**
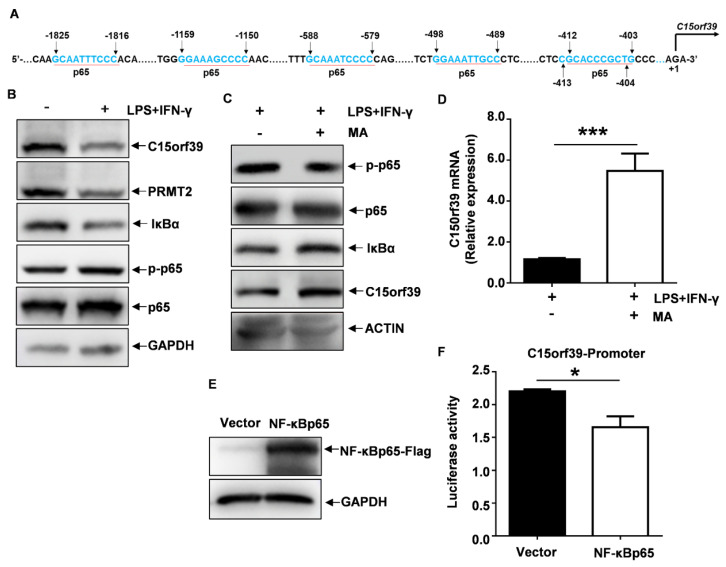
NF-κB signaling inhibits C15orf39 transcriptional expression. (**A**) Binding sites of transcription factors NF-κBp65 were predicted in the C15orf39 promoter using two online prediction systems [https://jaspar.elixir.no/ (accessed on 20 October 2016) and https://genome.ucsc.edu/ (accessed on 8 August 2003)]. (**B**) The protein levels of C15orf39, PRMT2, IκBα, and the phosphorylated and total NF-κBp65 were determined by western blot assay in HMC3 cells that were stimulated by LPS/IFN-γ for 24 h (*n* = 3). (**C**,**D**) HMC3 cells were pre-treated with maslinic acid (MA) for 3 h and then stimulated by LPS/IFN-γ for 24 h. The protein levels of C15orf39, IκBα, and phosphorylated and total NF-κBp65 (*n* = 3) (**C**) and the mRNA level of C15orf39 (**D**) were determined by western blotting and qPCR, respectively. (**E**,**F**) NF-κBp65-Flag or empty (vector) pcDNA3.1 plasmids, luciferase reporter vector pGL4.10/C15orf39 promoter (−2000 bp to +228 bp), and the control pRL-CMV-Renilla plasmid were co-transduced into HMC3 cells. After 48 h, NF-κBp65 expression was determined with anti-flag antibody using a western blot assay (*n* = 3) (**E**), and dual luciferase reporter gene expression is shown as the ratio of firefly to Renilla luciferase activity (**F**). (**D**,**F**) Data indicate means ± SD, *n* = 3, two tailed Student’s *t*-test, * *p* < 0.05, *** *p* < 0.001.

**Figure 8 ijms-25-06025-f008:**
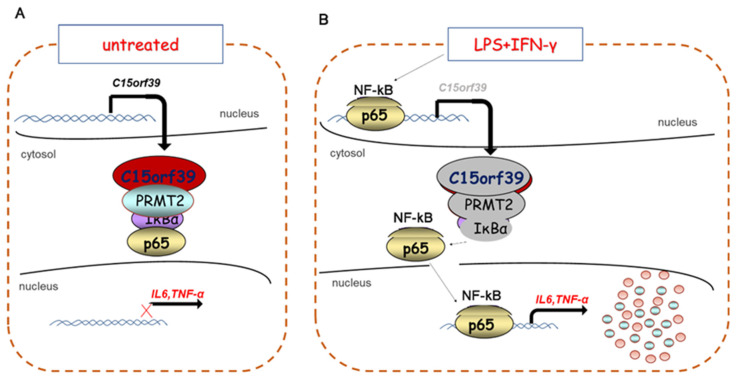
Schematic diagram of how C15orf39 suppresses IL-6 and TNFα expression in LPS/IFN-γ-stimulated HMC3. (**A**) Under steady-state conditionality (such as an un-treated condition), C15orf39 interacts with PRMT2 in the cytoplasm. The interaction of C15orf39 and PRMT2 enhanced the binding of IκBα and NF-κBp65, which inactivated NF-κB signaling. Thus, the inflammatory factors IL6 and TNFα were not expressed in the steady-state microglia cells. (**B**) Under activation conditions (such as simulation with LPS/IFN-γ), it was found that LPS/IFN-γ-activated NF-κB signaling suppresses C15orf39 expression. The binding of C15orf39 and PRMT2 declined in the cytoplasm, which led to a reduction in the binding of IκBα and NF-κBp65. NF-κBp65 was activated and induced NF-κB signaling activation. Thus, the inflammatory factors IL6 and TNFα appear to be induced in activated microglia cells.

## Data Availability

No datasets were generated or analyzed during the current study.

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
