# Peer review of "Human C15orf39 Inhibits Inflammatory Response via PRMT2 in Human Microglial HMC3 Cell Line"

_ijms, 2024, doi:10.3390/ijms25116025_

Round 1

Reviewer 1 Report

Comments and Suggestions for Authors

Human C15orf39 Inhibits Inflammatory Response via PRMT2 in Human Microglial HMC3 Cell Line

In this work the authors examined the inhibitory effects of human C15orf39 on inflammatory response in human microglial cells.

General revision: I suggest to check the English and the formatting of the figures.

Introduction: I suggest to highlight the importance of understanding the molecular mechanisms that regulate microglial inflammatory responses in the context of neurological disorders.

Matherials and methods: I suggest to add references for established methods or protocols used in the study.

Others: Have you tried to evaluate other pathways as NLRP3 or TLR? Have you evaluated other interleukins as IL-1?

Comments on the Quality of English Language

Moderate editing of English language required

Reviewer 2 Report

Comments and Suggestions for Authors

The manuscript " Human C15orf39 Inhibits Inflammatory Response via PRMT2 2 in Human Microglial HMC3 Cell Line” - is a search for the cause of the development of diseases related to microglia activation.

My comments:

1.       Introduction –

“Once inflammation response is excessive and uncontrolled, detrimental harm may appear in the host tissue [5, 7]” - in response to infectious and other factors, it creates pathological beta amyloid proteins.

“Thus, microglia may be regarded as a promising therapeutic target in nervous system diseases [11, 12, 13].” – on what basis?

A more complex process.

“C15orf39 is expressed in various tissues including the brain in the NCBI database.” –add Ref.

“suggesting that C15orf39 may be a potential therapeutic target in many central nervous system diseases.” - what neurological diseases”?

what was the purpose of the research?

2.       Methods –

Has consent been obtained for the research?

3.       Discussion –

“Thus, our study provides some valuable evidence for controlling the

detrimental role of NF-κB signaling in brain injury diseases by overexpressing C15orf39.” - the studies were conducted in vitro, they would need to be confirmed in vivo to suggest such a role.

4.       Conclusion -

“Our study provides new information on the molecular biological function of C15orf39. It also provides some valuable hints for the treatment of the central nervous system diseases by overexpressing C15orf39.” - such studies have not been conducted. Too forward-looking conclusions.
